# iPSC Technology: An Innovative Tool for Developing Clean Meat, Livestock, and Frozen Ark

**DOI:** 10.3390/ani12223187

**Published:** 2022-11-17

**Authors:** Rajneesh Verma, Younghyun Lee, Daniel F. Salamone

**Affiliations:** 1VG Biomed Thailand Ltd., 888 Polaris Tower, 6th Floor, Soi Sukhumvit 20, Bangkok 10110, Thailand; 2Laboratory of Reproductive Biotechnology, Building 454, Rm 343, Gyeongsang National University, 501 Jinjudae-ro, Jinju 52828, Republic of Korea; 3Department de Produccion Animal, Facultad de Agronomia, University of Buenos Aires, Av. San Martin 4453 Ciudad Autonoma de Buenos Aires, Buenos Aires B1406, Argentina

**Keywords:** IPSC, endangered, farming, Ark

## Abstract

**Simple Summary:**

Induced pluripotent stem cell (iPSC) technology is an emerging technique to reprogram somatic cells into iPSCs that have revolutionary benefits in the fields of drug discovery, cellular therapy, and personalized medicine. However, these applications are just the tip of the iceberg. There are examples now of repurposing iPSC technology to potentially aid endangered species, as well as even reviving extinct species. With increasing consumer reliance on animal products, combined with an exponentially growing population, there is a necessity to develop alternative approaches to conventional farming practices. One such approach that offers significant benefits is the development of domestic animal iPSCs, which will have a myriad of potential knock-on effects, such as reduction in animal death, pasture degradation, water consumption, and greenhouse gas emissions. Ultimately, reducing the environmental impact caused by large-scale farming provides an environmentally friendly commercial alternative. Another key issue that iPSC technology will address is both food security and potential zoonosis. Here, in the form of a “Frozen Ark”, we will discuss past, current, and future research using iPSC technology and how it will potentially impact protein production in the form of animal-free meat alternatives to address increasing public awareness of foodborne diseases, factory farming, and the meat industry’s ecological footprint.

**Abstract:**

Induced pluripotent stem cell (iPSC) technology is an emerging technique to reprogram somatic cells into iPSCs that have revolutionary benefits in the fields of drug discovery, cellular therapy, and personalized medicine. However, these applications are just the tip of an iceberg. Recently, iPSC technology has been shown to be useful in not only conserving the endangered species, but also the revival of extinct species. With increasing consumer reliance on animal products, combined with an ever-growing population, there is a necessity to develop alternative approaches to conventional farming practices. One such approach involves the development of domestic farm animal iPSCs. This approach provides several benefits in the form of reduced animal death, pasture degradation, water consumption, and greenhouse gas emissions. Hence, it is essentially an environmentally-friendly alternative to conventional farming. Additionally, this approach ensures decreased zoonotic outbreaks and a constant food supply. Here, we discuss the iPSC technology in the form of a “Frozen Ark”, along with its potential impact on spreading awareness of factory farming, foodborne disease, and the ecological footprint of the meat industry.

## 1. Introduction

Currently, we are experiencing the Holocene extinction or sixth mass extinction, which is directly due to human intervention [1,2]. Several groups around the world have tried to save the genetic blueprints of different animals in the form of cellular repositories or Frozen Arks. Such storage serves as the genetic capital to ensure the survival of endangered species as well as food production. The Frozen Ark’s ethos is to conserve knowledge before it is too late for future generations [2,3]. With time and technological advancements, the costs of preserving material and genome sequencing have declined over the past decade [1]. However, it is noteworthy that the Frozen Ark approach is not a substitute for preserving species but is a “Plan B” [1]. The Frozen Ark (www.frozenark.org accessed on 19 November 2015), established in 1996, aims at preserving the genetic material of endangered species. However, more recently, the “biobanks” have moved to preserve germplasm, tissue, blood, and DNA. Fertility preservation and reproductive technology are important tools for preserving endangered species and are closely linked to biobanking [1]. The integration of artificial insemination into conservation programs has been successful to some extent. Nevertheless, no wild species is currently being preserved using embryo-based or oocyte cryopreservation approaches [4], which might be attributed to inadequate knowledge regarding the species biology, expertise, facilities, and necessary funding for successful implementation [3].

The main aims of the Frozen Ark approach are as follows [3]:To provide a database of stored and accessible specimens.To enhance sample collection, processing, conservation, and distribution.To make biological material usable for conservation programs to help counter genetic erosion.To safeguard valuable genetic material for scientific research, advancing awareness, and benefiting humans.To disseminate information on the current global extinction crisis, its impact on genetic biodiversity across the planet, and the impact of genetic management of endangered species in their fight for survival.

We are currently building a comprehensive Frozen Farmyard library of fibroblasts [5]. These fibroblasts will ultimately serve as a cellular source for induced pluripotent stem cell (iPSC) lines using the non-integrative mRNA method [6]. The availability of cell lines from different species and various breeds within species will enable the cell-based meat industry to conduct insightful, basic research [5,7]. The creation of these critical starting materials will open the door to cell-based meat research and other exciting avenues of research.

## 2. PSCs in Livestock and Wildlife

In both mice and humans, embryonic stem cell lines (ESCs) have been established. However, this is not the case for farm and wild animals [8]. The emergence of iPSC technology in 2006 (Figure 1) offered an alternative Pluripotent Stem Cells (PSCs) generation approach that can be translated to both farm and exotic animals [9,10,11,12,13,14]. Nuclear transfer reprogramming was the first stream. John Gurdon stated in 1962 that his lab had produced tadpoles from unfertilized eggs with a nucleus made from adult frogs’ intestinal cells. Ian Wilmut and colleagues announced the creation of Dolly, the first mammal created through somatic cloning of mammary epithelial cells, more than three decades later. These somatic cloning successes showed that somatic cell nuclei may be reprogrammed in oocytes and that even differentiated cells have all the genetic material needed for generating whole animals. Takashi Tada’s team demonstrated, in 2001, that ESCs also include elements that can reprogram somatic cells.

The identification of “master” transcription factors made the second stream. In 1987, it was discovered that the Drosophila transcription factor Antennapedia, when produced ectopically, causes the development of legs rather than antennae. In the same year, it was discovered that the mammalian transcription factor MyoD transformed fibroblasts into myocytes. The idea of a “master regulator”, a transcription factor that decides and influences the fate of certain lineages, was developed because of these findings. Several scientists have started looking for lone master regulators for different lineages. With a few exceptions, these attempts were unsuccessful.

The study of ESCs is the third and most significant research area. Austin Smith and others have developed culture conditions that permit the long-term maintenance of pluripotency since the first generation of mouse ESCs in 1981. Leukemia inhibitory factors are essential for maintaining mouse ESCs (LIF). Like this, the ideal culture conditions with basic fibroblast growth factor (bFGF) have been established from the first generation of human ESCs.

By combining the findings from the first two lines of inquiry, we could develop a hypothesis and develop experiments to test it: Somatic cells in oocytes or ESCs can be reprogrammed to become embryonic cells by a complex interaction with several variables [15]. We could then pinpoint four parameters that can produce iPSCs using knowledge of the culture conditions required to culture pluripotent cells [15].

Capable of being derived from various types of accessible somatic cells, iPSCs offer an ethically acceptable and endless source of PSCs [16,17,18,19,20,21,22,23].

The first iPSCs were generated via retroviral transduction of Sox2, Oct4, c-Myc, and Klf4 into a donor cell genome [9,10]. However, recently, there have been attempts to develop non-integrating approaches that generate “clean iPSCs” with a pristine genome [12]. These approaches include viral delivery of RNA via the Sendai virus, transfection of modified mRNA [6], self-replicating mRNAs [6], episomal approaches, and protein-based reprogramming.

iPSC technology potentially acts as a major player in aiding with environmental protection and enhancing animal conservation [24]. This technology could provide a safety net to save current and future endangered species, or in the worst-case scenarios, could aid in de-extinction [19,20,21,22,23]. Currently, with limited funding, it must be considered that the iPSC technology may have broader applications. Manufacturing clean meat from iPSCs derived from domestic animals, such as cows and pigs [25,26], could reduce the environmental impact of commercial animal husbandry [12,13]. One can also think of the exploitation of iPSCs to obtain exotic animal products without harming the animals (Figure 2) [2]. For example, rhino horn or ivory that are produced in vitro could essentially compete with their black-market counterparts, which might lead to a reduction in poaching, and, in turn, protect the extinction of already endangered species [2]. The potential of iPSC technology with respect to the conservation of species and the environment is limited only by our creativity and ambition. Here, we discuss the iPSC generation technology for mammalian farm animals [11,12,13,17], and discuss its potential applications in ensuring the supply of clean meat and prevention of species extinction [24].

### 2.1. iPSCs, Bioreactors, and Bioprinting

We are now entering the “Holocene extinction” or “sixth mass extinction” as several species are being destroyed owing to human intervention. It has been reported that about 150–200 species of birds, mammals, insects, and plants become extinct each day [2]. Human activities and urbanization also require land, which leads to a reduction in the forest areas and irreversible damage to the forest ecosystems. The Food and Agriculture Organization of the United Nations (FAO) has reported that livestock are the main consumers of global land resources, earmarking about 80% of agricultural land [27], which leads to greenhouse gas emissions and deforestation [27]. In addition, the consistent use of antibiotics in animal feed has led to the development of antibiotic-resistant microbial strains, which further harm the human population [28]. Such environmental and health risks have shifted the consumers’ interests toward more environmentally sustainable animal products. One of the techniques to produce such products is known as “cellular agriculture”, which involves stem cell research. This technique is aimed at creating the animal-based products in vitro. More importantly, it does not involve harming or killing animals and potentially reduces the farming footprint in terms of land use and environmental impact. Animal stem cells are extracted via biopsy and replicated in vitro, followed by adequate modification to obtain the desired animal products. The desired cellular farming protocols have already been devised for a range of farm animals, facilitating the exploration of laboratory-generated animal products [29].

It is noteworthy that cellular farming has been practiced for several decades. There have been several attempts previously to produce animal products without the use of live animals, such as recombinant proteins like insulin and rennet. However, cellular agriculture was first used to produce meat in 2012 by Mark Post and his team [30]. Their protocol required three months to generate the adequate quantity of muscle cells sufficient to produce a burger. However, the production cost of the burger was extremely high; the cost of the 85 g burger was $325,000 [31]. Nevertheless, their research provided a proof of concept, prompting many commercial firms to embark on lab-grown meat products [32].

The concept of cellular farming has been employed on other types of meats too, such as pork [31,33,34,35,36,37,38,39,40,41,42,43,44,45,46,47,48]. Genovese and colleagues devised a protocol efficient skeletal muscle derivation from pig iPSCs. While this technique was entirely in vitro, the derivation of iPSCs still required cells from an animal source; thus, the product was not completely “animal-free” [31]. The in vitro culturing of iPSCs under serum-free conditions and in the absence of other animal products remains a major challenge. For in vitro cellular proliferation, most protocols require the use of animal products, such as fetal bovine serum (FBS), serum-derived products, or extracellular matrix [49]. The regulatory agencies demand the production of cells and any future iPSC meat or consumer products under xeno-free conditions [41]. Certain protocols have been devised for a feeder-free and xeno-free stem cell culturing to either eliminate or reduce the use of animal products in compliance with the regulatory constraints and to improve quality control processes [43,44,45]. Such media will not only eliminate the use of animal proteins, but also antibiotics and hormones.

Currently, cellular agriculture is highly costly [50,51,52,53,54,55,56,57,58,59,60,61,62,63,64,65,66]. It is estimated that the current cost of laboratory meat is ~$40,000 per kg, making it a very exclusive product. To reduce the cost, improved approaches, such as mass cell culture, are warranted.

Bioreactors facilitate suspension culture capable of producing abundant iPSCs and their derivatives within a few days [56]. Previous studies have demonstrated high mouse iPSC proliferation using stirred bioreactors [67] and scaling of human iPSCs using xeno-free media in bioreactors [67]. Furthermore, it is possible to collect and combine animal iPSCs from a bioreactor to mimic the actual processed meat product, which significantly lowers the final product cost than that obtained using conventional cell culture techniques [67].

In addition, to obtain highly ordered complex tissue, it is also necessary to insert the cells collected from an animal into a scaffold with specific vascularization and porosity [30]. While the generation of such complex tissues with micro vascularization is difficult, this problem may be overcome using the three-dimensional (3D) “bioprinting”. The process of hydrogel employs living cells that are suspended in hydrogel; this suspension can then be polymerized in the form of any complex 3D structure using computer-generated models [50,56,68]. A previous study has also demonstrated the generation of artificial skin constructs using human iPSCs inserted in alginate hydrogel [36]. The animal iPSC-derived fur and skin derived could be used as an alternative for natural fur and leather, which would be especially beneficial in the case of exotic animals, such as crocodiles [68,69,70,71,72,73,74,75,76]. To increase public awareness in this context, it is necessary to advertise the naturally obtained animal products, which, in turn, would help generate higher incentives for research on cellular farming and increase the demand for environmentally sustainable products [73,74].

The initial cell sample is fed with the nutrients and water required to grow and replicate [48]. Later, the cells are induced to differentiate into muscle, fat, and connective tissue that constitute meat. A support system (or scaffold) was then introduced to provide the cells with instructions on how to organize themselves into the correct 3D structure [68]. This whole process can be conducted in a grower (also known as a bioreactor) (Figure 3) [7].

### 2.2. Why Clean Meat?

The meat developed in vitro is termed as clean meat and has been referred to as a potential substitute for the conventional meat [31,32,33,34,35,36,37,38,39,40,41,42,43,44,45,46,47]. Traditional animal products are said to be unsustainable because the live source animals consume a large amount of feed, of which most of the generated energy is wasted by the animal for daily activities and the production of non-edible tissues [44]. Compared to the plant-based industries, the animal-based industries exhibit a more severe environmental footprint, especially in the context of water and land usage and greenhouse gas emission, with the worst environmental impact exhibited by the beef industry [44,45].

Initially, some investigators reported that chicken muscles could grow efficiently in the absence of live chickens [46,48,77]. Since then, many researchers have explored the possibility of producing meat in vitro. For the last 15 years, skeletal muscle stem cells have been used to generate cultured muscles for potential medical applications [47]. In another study, NASA used turkey cells to produce muscle culture and goldfish cells to produce the first edible lab-grown fish filet. Their study demonstrated that muscle strips could be produced by introducing a collagen matrix into the stem cell culture [78]. The emergence of the meat cultivation consortium led to the first meat cultivation symposium at the Norwegian Food Research Institute in Norway in 2008 for the exploration of potential applications of lab-grown muscle tissue [51]. Other studies have devised protocols to produce bone, skeletal muscle, fat, fibrous tissue, and cartilage [52,53,54,55,56,57,58,59]. Lab-grown meat, derived from the bovine stem cells, was first used to make a burger in 2013; however, the meat itself was very costly and requires around 10,000 individual muscle strips to mimic the natural product [60,61,62,63,64,65,66]. Even with the current progress, many puzzles still need to be solved to obtain the optimum meat substitutes for the general population using feasible methods [68,69,70,71,72,73,74,75,76,77,79,80,81,82,83,84,85,86].

## 3. Clean Meat Production

The idea of artificially creating meat can be traced back as far as 80 years, when Frederick Edwin Smith predicted that “to eat his steak, it will no longer be necessary to go to the extravagant length of rearing a bullock. It will be possible to grow as large and as juicy a steak as desired from one parent steak of tenderness of choice.” [77,80,81,82,83,84,85,86]. In the 1930s, Winston Churchill commented on “the absurdity of growing a whole chicken to eat the breast or wing by growing these parts separately under an appropriate medium” [57,58,59,60,61,62,63,64,65,66,68,69,70,71]. Since then, for the realization of the idea, two major technologies have been developed. The basic principle is the use of a biotechnological approach that broadly involves cell culture and tissue culture/tissue engineering techniques, technically known as ‘scaffold-based’ and ‘self-organizing’ techniques [77,87,88].

### 3.1. Self-Organization Technique

The first technique involves the use of a donor animal’s muscle explant, which proliferates in a nutrient medium [66,68,69,70,71,72,73,74,75]. Previously, Alexis Carrel was successful in keeping a piece of chick heart muscle alive and beating in a Petri dish, demonstrating the possibility of keeping the muscle tissue alive outside the body in the presence of sufficient nutrients [77]. However, the actual concept was conceived of in the early 21st century, with the use of tissue-engineering techniques to produce meat [71,72,73,74,75,76,77,79,80,81,82,83,84,85,86,87,88]. Previously, the researchers placed skeletal muscle explants from goldfish (*Carassius auratus*) in various culture media, demonstrated a varied pattern of growth over a period of seven days [30,65]. The explants were also placed in a culture containing dissociated skeletal muscle cells of Carassius, resulting in a 79 percent increase in the explant surface area [52,53,54,55,56,57,58]. The muscle tissue has also been kept alive in a fungal medium; in such a medium, the chicken muscles could be preserved for up to two months [66]. Despite such successful attempts and advancements, the lack of blood circulation still poses a major hurdle to long-term success [66]. The explant method can be applied to the in vitro Meat Production System (IMPS) since the researchers suggested that the produced tissue formed would closely resemble meat [63]. Nevertheless, the problem remains, that is, the limited proliferation potential; the donor animals still require new biopsies on a regular basis [64].

Self-organization helps to create standardized food; the cultivated meat would have a well-defined 3D structure, at par with natural meat conformation. The same result can be achieved using the tissue engineering principles for de novo muscle tissue synthesis [57,68].

### 3.2. Scaffold-Based Technique

Another meat cultivation method is the scaffold-based technique, which involves the use of an abundant number of stem cells obtained from several tissues. Here, the embryonic myoblasts or adult skeletal muscle satellite cells are proliferated, bound with a carrier or scaffold, and then cultured in a bioreactor [30,50,55,65,67,81].

It works on the principle that, when suspended in a bioreactor and grown on a scaffold, one can obtain a large quantity of muscle cells. These cells form myofibers, which are then harvested, processed, and consumed [50,56,65,66]. Currently, two protocols have been proposed for meat cultivation in vitro. Both protocols are similar, but separately devised by Vladimir Mironov and Willem van Eelen for NASA [56]. Vladimir Mironov proposed the use of collagen spheres to which cells can be bound and then grown in a bioreactor [56]. In contrast, Willem van Eelen proposed the use of a collagen meshwork, along with an occasional change of fresh culture medium. Alternatively, the culture medium can be percolated through the meshwork, while collagen can be replaced by other artificial substrates or edible proteins. It can also use 2D sandwiched myocyte monolayers after harvesting. This technique is ideal for processed ground meat products, but not for highly structured meat products, such as steak. Structured muscle tissue requires a more refined approach, such as the self-organizing technique [56].

### 3.3. Tissue Engineering Techniques

Vladimir Mironov proposed another approach using tissue engineering techniques that could be used to develop an artificial muscle [65]. He proposed a list of the polymeric substances that could be used for cell attachment and nutrient perfusion as well as the concept of co-culturing myoblasts with other cell types to obtain a product that mimics the actual muscle structure. However, there is limited information on the creation of artificial capillaries [65].

#### 3.3.1. Lab-Grown Meat—Potential Benefits

Compared to its conventional counterpart, lab-grown meat could result in reduced greenhouse gas emission, water use, eutrophication, and land use [30]. However, they found that cultivated meat exhibited the highest impact across several environmental categories, mainly owing to its high energy requirements, except for land use and freshwater and oil ecotoxicity [30]. Previous studies have also shown that lab-grown meat exhibits a lower environmental impact than beef and pork, but a higher environmental impact than plant-based proteins and chickens [60,62]. However, all Life Cycle Assessments reported significant potential of cultivated meat that could lead to better environmental outcomes than those shown by current models [62].

In addition, cultivated meat exhibits a lower potential to being infected with pathogens, owing to its sterile growing conditions, which facilitates better food safety and quality [57,60]. However, many aspects of cultivated meat still need to be studied, such as genetic instability due to multiple cell divisions and variable media components; a full analysis of product traceability could ensure science transparency [78].

The production of cultivated meat employs a significantly smaller number of animals compared to conventional farming [30,57,60,62]. This could be appealing from an animal protection viewpoint for vegetarians, vegans, and the omnivores who are trying to reduce meat consumption on moral grounds [53]. While the economic value of cultivated cells is still unknown, the harvesting of large numbers of cells from lesser animals indicates higher yields per animal [74]. Such profitability makes in vitro farming a viable alternative to traditional farming systems, including the Concentrated Animal Feeding Operations (CAFO) [75].

Cultivated meat could also prove to be a new opportunity for those using native livestock breeds [79]. The shift from carcass to cell harvesting might lead to a shift from the tedious selection protocols of high-yielding livestock to more traditional livestock that would otherwise require low-input, extensive, and low-density systems [81,82,83,84,89]. Thus, cultivated products exhibit three advantages: reduced environmental impact due to low impact systems, high profitability, and the preservation of conventional breed genetics and biodiversity [80,81].

Traditional carcass utilization in the commercial meat industry is the single biggest challenge with respect to waste management. Cultured meat provides a novel opportunity by producing only the primary cut that is consumed or processed entirely [76,79,80,81,82,83,84].

The producer of cultivated products can also create customized versions of the product (such as craft brewers, farm cheesemakers and charcuterie producers now). Thus, the cultivated products offer them flexibility as well as competition in the market, along with a higher skilled employment [73,74,75,76]. If sustained, the combination of conventional agriculture and new technologies might facilitate the generation of a circular economy, since most of the waste materials (heat, metabolites) produced during the meat cultivation can be used on a farm [81]. It is also possible to establish a true cost accounting structure to realize the financial and environmental impact of cellular farming [72,73,74,75].

#### 3.3.2. Lab-Grown Meat—Technical Challenges

The major challenge in cultivating in vitro meat is to mimic the muscle growing environment found inside an animal body [85]. Like any other tissue, muscle tissue engineering involves combined knowledge of the mechanism of tissue development and growth and biochemical engineering for appropriate replication of the in vivo environment [54,65,82]. To date, the major applications of tissue engineering have been in the medical field, such as regenerative medicine and drug discovery. For the processing of cultivated food, the basic concepts are the same, except for larger quantities and lower cost. When considered as a food item, the lab-grown meat can be produced within less stringent regulatory guidelines and has lower quality compared to the meat used for medicinal purposes [79,81,82,85,86].

Lab-gown meat can be derived from skeletal muscle stem cells (myo-satellite cells) that are extracted from live animals and induced to grow and differentiate in vitro; however, there is no commercially viable manner for conducting such a process.Cultured meat is highly advantageous over traditional meat, especially in terms of resource consumption, greenhouse gas emission, animal welfare, and variation in the nutritional composition. Nevertheless, customers may be skeptical about embracing these goods because of “un-naturality” or “artificiality” beliefs.The culture media for the growth of cultivated meat must consist of appropriate levels of oxygen, nutrients, bioactive compounds, and growth factors and bioactive compounds necessary for normal muscle production. Certain media components are still obtained from animal sources, and it is desirable to get rid of these components to make the media entirely animal-free.Previous researchers have already been able to obtain myocyte monolayer and muscle fibers that are centered on collagen fibers. However, a steak-like 3D structure would involve the use of a scaffold, while ensuring a continuous oxygen and nutrient supply, and elimination of waste products, such as CO_2_It is possible that the initial cultivated meat products would mimic processed meat items with minimal structural requirements. Nevertheless, steak- or roast-like products likely require a more developmental work on scaffolds, circulatory systems, and key performance attributes, including tenderness and flavor.Cultivated meat products can play a useful complementary role in meeting the expected increase in global demand for meat. It remains to be seen to what degree they rival their traditional counterparts.It is still too early to assess the ability of consumers to consider the consumption of meat products cultivated in the laboratory.

#### 3.3.3. Transgenesis for Large Mammals

Transgenic farm animals have become increasingly popular over the past few years as excellent human disease models (Figure 4) [89,90,91,92]. The availability of annotated genome depositories and active transgenesis methods dramatically increased the success rates of transgenesis in farm animals [24,90,91,92,93,94]. Today, the molecular tool repertoire facilitates the precise and rapid modification of large mammalian genomes. The development of iPSC technology could contribute to these advances by facilitating desired genetic modifications via in vitro high-performance screens, followed by SCNT or blastocyst complementation for successful generation of transgenic offspring [95,96,97]. Strengthened efforts are therefore needed to assess the potential of such iPSC lines in germ line contribution and chimera formation [89,97]. In addition, the potential of auxiliary small molecular inhibitors of the stemness signaling pathways in livestock iPS cells is already known [98]. Such small molecules might require high-performance screens for their identification. Recent advances in farm animal genetic engineering enable precise genetic modifications, which could be instrumental in advancing the preclinical testing of novel therapies [25,91,92,97]. To date, iPSCs have been derived from cattle, dog, rabbit, goat, horse, pig, and sheep; these cells have been shown to exhibit typical pluripotency characteristics, including teratoma formation and in vivo differentiation [89,90,97]. However, most of these iPSC lines have not been evaluated for chimera formation. A recent study demonstrated the use of porcupine iPSCs for chimera formation using blastocyst supplementation [12]. Similarly, when into eight-cell stage embryos, ovine iPS cells made a moderate contribution in the formation of chimeric lambs [95]. These experiments are an important step toward elucidating the mechanistic nature of iPSC pluripotency in farm animals. Thus, iPSC technology can become instrumental for advanced transgenesis in large mammals [24,95].

#### 3.3.4. Preservation of Endangered Species and Genetic Resources

Cryopreservation is an important and useful approach to preserve endangered wild and domestic species as well as their genetic material [20,21,22,23]. Several studies have shown the potential of iPSCs to prevent the extinction of several valuable species, such as the snow leopard, Bengal tiger, drill monkey, and white rhinoceros [8,19,21,23,99]. Some studies have also shown the potential of iPSCs to be used as SCNT donor cells or as genetic material banks [4,20]. The iPSCs may also be used to generate mature spermatozoa and oocytes, which may then be used to produce in vitro embryos. However, the livestock iPSCs have not yet been used to produce functional gametes in vitro.

#### 3.3.5. Applications of iPSCs

(a)Saving the endangered species

The Generation of iPSC lines from adult tissue makes it possible to add a critically important safety net to preserve endangered animal species [19,20,22,23,99,100]. Nevertheless, iPSCs still need to prove their efficiency in assisted reproductive technologies (ART). The first example of iPSC-derived fully functional animal was demonstrated for mice with the aid of tetraploid complementation assay, in which an in vitro tetraploid blastocyst was first injected with iPSCs, and it was then implanted into adult female mice [10,11]. Later, similar iPSC lines were developed for other species, such as bovines, primates, birds, etc. [2,4,6,9,11,12,13,14,16,17,18,19,20,21,22,99,101].

The inability to generate stable transgene-free iPSCs makes it difficult to derive these cells from large animals [24]. Hence, it is necessary to further study species-specific reprogramming factors that could facilitate understand the missing links for efficient iPSC development and sustainable offspring generation [2,99]. Bovine adult fibroblasts were only produced when nanog, SV40largeT, and KDM4A were expressed in combination with the other reprogramming ingredients [102]. By utilizing a specific component such as SV 40 large T, Lin28, and hTERT sheep somatic cells can be directly transformed to become induced pluripotent stem cell [103,104]. Another study reported the creation of live piglets using the iPSC technology in 2013 [13]. In this study, the iPSCs (nuclear donors) were combined with enucleated donor eggs. It has also been reported that both intracellular nuclear injection and tetraploid complementation could be used to produce viable offspring. However, such approaches have still not been applied for any endangered species [4]. This finding may be attributed to the unavailability of donor eggs or embryos for such species. Therefore, producing embryos from iPSC-derived gametes would be the only way forward. Fully functional iPSC-derived spermatozoa and oocytes have been documented in mice; however, such immature gametes had to be injected into an adult mouse ovary to make them functional [2,11,14].

Recently, ESCs and embryos were generated from the critically endangered northern white rhinoceros (NWR, *Ceratotherium simum cottoni*) [8]. In March 2018, the last male member of this species died, leaving only two infertile females [99]. To prevent their extinction, oocytes were obtained from a related subspecies, southern white rhinoceros (SWR, *Ceratotherium simum simum*), which is not currently at risk [99]. They were then matured and fertilized via intracytoplasmic sperm; the fertilized eggs were grown until the blastocyst stage. These hybrid SWR–NWR embryos were then implanted in a female SWR, while preserving NWR genes using ART. Next, the NWR oocytes were generated from cryopreserved somatic NWR tissue using the iPSC technique. Thus, in cases with limited availability of genetic material, iPSCs could be used to retain the genetic diversity and raise the size of a critically endangered species population (Figure 5) [99].

Based on the rapid technological advancements, iPSC technology may be used to develop mature, fully functional gametes, in the absence of any extraneous tissue, to form viable embryos for other extinct or endangered animals [26]. However, the generation of such embryos in vitro would still require the resolution of a plethora of ethical issues. Most of the research on animal stem cell-based reproductive techniques has revealed a viability rate of only 5%–13% [92,95]. Furthermore, there has been limited research on the regulatory mechanisms of pluripotent iPSC generation. iPSC reprogramming factors need to be further elucidated to help facilitate the restoration and preservation of wildlife. The key goal of wildlife conservation is to prevent the extinction of species; iPSCs and genetic engineering can potentially be used to reverse any damage that was once considered irreparable [2,20]. However, the animals obtained using these techniques would have to be raised in captivity, unable to be released into the wild. Another challenge that ensues the resurrection of any species is the uncertain impact of its behavior on the ecosystem [101].

(b)Tissue and disease Research in domestic animals

iPSCs can be useful tools for modeling tissues and illnesses. The study of tissue diseases and developmental processes has been made possible by the in vitro differentiation of iPSCs, which may also enable the preclinical testing of medicinal medications for veterinary use.

In mouse iPSC research, differentiated iPSC lines have been successfully used to simulate disease and high-throughput screening of small compounds for their effects on disease development. This method eliminates the need for interspecies comparisons or overuse of lab animals when evaluating prospective treatments against disease-genotype cells unique to an individual or species. Numerous instances of differentiation into certain cell types in porcine, equine, canine, galline, and bovine models have been documented.

The degree of differentiation varies from progenitor cells to completely differentiated cell types, even though the characterization of these differentiation cells was proved by the physiological, genetic, or metabolic characteristics of cell lines. Consumers of farmed animal by-products are at risk of contracting domestic animal illnesses, which are widespread. Unfortunately, the use of stem cells in animal disease research is new and currently scarce. The extended self-renewing property of iPSCs enables their application in the research of domestic species physiology, disease pathology, medication toxicity, and vaccine development.

The use of induced pluripotent stem cells is a cutting-edge technique with enormous promises for advancing veterinary care. Tissues that can be examined for their physiological functions and disease pathologies can now be produced thanks to iPSC cultures. Additionally, iPSCs themselves could one day be used to treat various ailments that veterinarians see [105].

#### 3.3.6. iPSCs in Preclinical Studies

The iPSC technology has opened new perspectives in elucidating the onset and progression of diseases, pluripotency, and regenerative medicine [92]. However, the challenges of lower growth kinetics and the efficiency of iPSCs silenced ectopic transgene reactivation, insertional mutagenesis risk, and potential tumor formation must be resolved before introducing iPSC-derived therapies for clinical application [92]. The biosafety of transplanted variants of iPS cells is a significant aspect [106]. Several studies have indicated that iPSCs could harbor epigenetic mutations, genetic mutations, and variations in copy numbers [49,67,100]. Such aberrant changes can alter the tumorigenicity of iPSCs and iPSC-derived cells. Retro- and lenti-viruses that are commonly used for the introduction of reprogramming factors into differentiated cells may increase the immunogenicity of farm animals. They also serve as model organisms to assess the risks and obstacles in pre-clinical longitudinal studies [24,90,91,92]. Compared to rodent models, iPSCs are more analogous to humans in terms of life span, anatomy, metabolism, and pathophysiology [14]. Mammalian models could help optimize the cell doses required to achieve adequate therapeutic effects and elucidate the fate and functional integration of transplanted cells in the host tissue [24,92]. Farm animal-derived iPSCs could thus help human patients to develop novel cell therapies. Apart from the genetic disorders, iPSC technology can also be used for treating somatic diseases that affect a wider population. Several iPSC-based agents have been employed for the treatment of rare, monogenetic disorders [92]. Hence, elucidation of genetic disorder targets could prove to be beneficial for many patients. Notably an iPSC-based drug screening approach is still in its infancy [92,100].

## 4. Conclusions and Future Prospects

The research on the creation of iPSC from farm animals using their fibroblast repository was summarized in this review. Despite its unquestionable potential in the fields of agriculture and biotechnology, iPSC research has still not received the deserved attention. Approximately thousands of studies are currently focusing on murine and human cell reprogramming, and we could find many studies that described cell reprogramming in important mammalian farm animals [24,89,94,97,101,107,108,109,110,111,112,113,114,115,116,117,118,119]. We propose that future studies should focus on the agricultural and biopharmaceutical applications of the iPSC technology. We also urge that future studies on lab-grown meat using iPSCs should explore the mRNA non-integrative approach. Furthermore, iPSC technology can save the exotic species in the form of an SCNT donor source and convert them into in vitro gametes.

## Figures and Tables

**Figure 1 animals-12-03187-f001:**
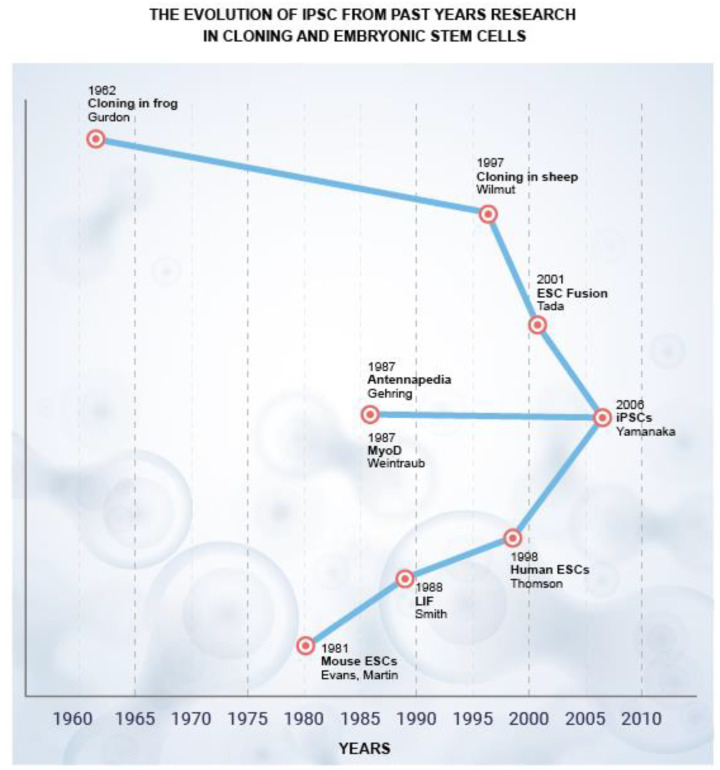
Pictorial presentation of the evolution of iPSC in 2006 from the past years of research on cloning and embryonic stem cells.

**Figure 2 animals-12-03187-f002:**
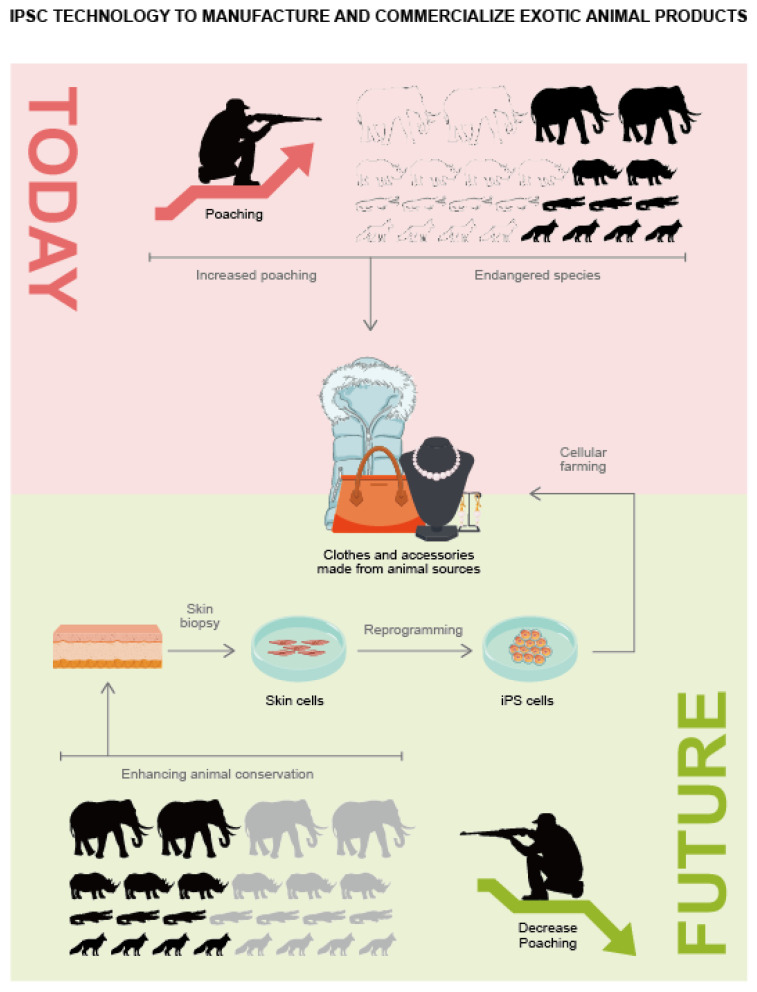
The use of iPSC technology for the production of exotic animal products could decrease animal poaching.

**Figure 3 animals-12-03187-f003:**
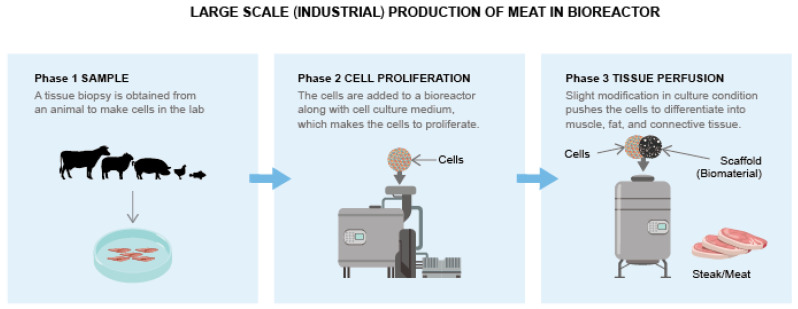
The production of meat in a bioreactor at the industrial scale.

**Figure 4 animals-12-03187-f004:**
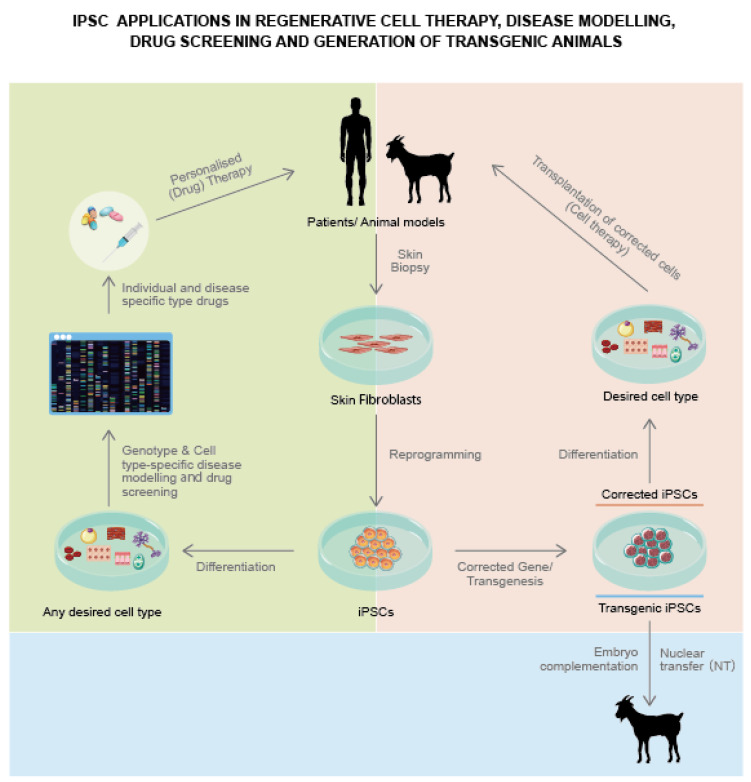
Most promising applications of iPSCs.

**Figure 5 animals-12-03187-f005:**
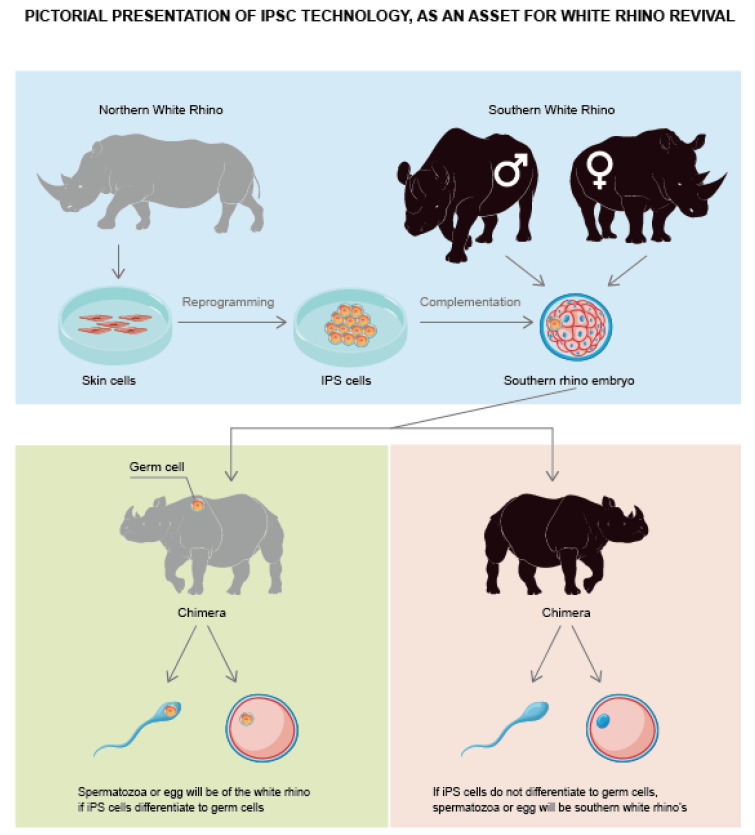
Pictorial presentation of iPSC technology as an asset for White Rhino revival.

## Data Availability

Not applicable.

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
