# Peer review of "iPSC Technology: An Innovative Tool for Developing Clean Meat, Livestock, and Frozen Ark"

_animals, 2022, doi:10.3390/ani12223187_

Round 1

Reviewer 1 Report

This review article introduces the lately noticed new potential of stem cell technologies easily and briefly.  This will contribute to the exploration and further development of stem cell technologies as well as the promotion of cross-disciplinary researches.  Thus, this article is considered significant as a review article; however some improvements are required as below.

1.  In the ‘Simple Summary’ on the top page, the Journal Instruction to the Authors instead of the manuscript summary is described.

2.  There are problems with the references throughout the manuscript. For example, the references are not displayed in numerical order (like 1-22-2-12-37-11). The reference number should be corrected. Furthermore, more importantly, the references described in the Manuscript do not match properly to those in the References later (like ref. 37, 38, 39, 40 on page 4-5). The authors should check and certify all references.

3. The Figure 1 is not described in the manuscript. It should be mentioned.

4. Spelling and grammatical errors are scattered. Please check the manuscript thoroughly.

5. Although the authors focused on the upcoming applications of stem cell technologies from the point of view of farm animals, they do not point out these potential usages in veterinary. This aspect should be described in the ‘Applications of iPSCs.’

Author Response

Review 1

This review article introduces the lately noticed new potential of stem cell technologies easily and briefly.  This will contribute to the exploration and further development of stem cell technologies as well as the promotion of cross-disciplinary research. Thus, this article is considered significant as a review article; however, some improvements are required as below.

  1. In the ‘Simple Summary’ on the top page, the Journal Instruction to the Authors instead of the manuscript summary is described.

Response: As suggested by the reviewer we have revised the Simple summary in the manuscript. Please see lines 18-32

  1. There are problems with the references throughout the manuscript. For example, the references are not displayed in numerical order (like 1-22-2-12-37-11). The reference number should be corrected. Furthermore, more importantly, the references described in the Manuscript do not match properly to those in the References later (like ref. 37, 38, 39, 40 on page 4-5). The authors should check and certify all references.

Response: We thank reviewer to identify the reference order and its numbering. Therefore, we have amended the references accordingly in the whole context. Please see lines in the context as mentioned 97, 126, 139, 149, 185, 196, 198-199, 218-219, 233-234, 251-252 254-255, 256-257, 262-263, 265, 269, 276-277, 279, 281, 344, 349-351, 355, 360, 401, 427, 438, 444-445,

  1. Manuscript do not match reference fixation of 37, 38, 39, 40.

Response: We have changed the reference completely 37 to [101] please see line 163 in the manuscript

[38] reference details changed, please see line 165

[39] reference changed to [102]. Please see line 174

[40] reference changed to [103] Please see line179

  1. The Figure 1 is not described in the manuscript. It should be mentioned.

Response: We have described the Figure 1 in the manuscript, please see lines 98-124

  1. Spelling and grammatical errors are scattered. Please check the manuscript thoroughly.

 Response: We have checked all grammatical errors throughout the manuscript and amended carefully. Please see lines as listed from 47, 49, 52-53, 55, 59, 60-61, 66, 77-78, 87, 89, 91-92, 95, 125-126, 141-144, 147-148, 150, 161, 169, 171, 173, 176, 178, 188-190, 192, 194, 197, 204, 212, 217, 223-225, 232, 234-236, 243, 248, 251, 253, 255-256, 259, 263, 267, 271-275, 279, 284, 290, 295, 298, 301, 305, 311, 316, 323, 325, 328, 333, 345, 353, 355, 357, 363, 368, 370, 374, 376, 382, 384, 390, 392-393, 396, 400, 402, 420, 421, 429, 433, 437, 439, 442, 444, 449, 451, 457, 461, 463-464, 466, 476-477, 479, 482, 485, 486, 516, 518, 523, 528, 535, 536, 539, 548, 550, 556.

  1. Although the authors focused on the upcoming applications of stem cell technologies from the point of view of farm animals, they do not point out these potential usages in veterinary. This aspect should be described in the ‘Applications of iPSCs.’

Response: We have added about potential usages in veterinary area of science in the manuscript. Please see lines 489-514

Reviewer 2 Report

The review article describes the use of induced pluripotent reprogramming technology in conservation/preservation approaches and also the production of lab meat. The paper, in general, is well written with minor errors. However, there are some important corrections to be made to the review including the adding recent developments in the field before this is ready for publication.

Below are my comments:

1.    Line 345 ..."However, the re- 345 search on iPSC cultures of several domestic animals is still in its infancy". Please refrain from making such strong statements and please tone it down. It is true that domestic animal reprogramming advances have not reached the level to that of murine or primate, but in the recent years there has been several advances in domestic animal reprogramming technology. Please see references included below.

Su, Y., Wang, L., Fan, Z., Liu, Y., Zhu, J., Kaback, D., ... & Tang, Y. (2021). Establishment of Bovine-Induced Pluripotent Stem Cells. International journal of molecular sciences22(19), 10489.

Pillai, V. V., Koganti, P. P., Kei, T. G., Gurung, S., Butler, W. R., & Selvaraj, V. (2021). Efficient induction and sustenance of pluripotent stem cells from bovine somatic cells. Biology open10(10), bio058756.

2.  Line 347:  Several papers published have indeed shown the potential of ipsc to differentiate into the 3 germ layers.  If germline contribution was what was intended, it is confusing to the reader as to what germ cell differentiation means. In short, if the author intended to convey was about contribution to germline, please rephrase the sentence.

Line 348: The statement on use of small molecules on iPSC derivation in livestock has been examined and also been reported. Please correct the sentence and include the below references. 

Xu, J., Yu, L., Guo, J., Xiang, J., Zheng, Z., Gao, D., ... & Han, J. (2019). Generation of pig induced pluripotent stem cells using an extended pluripotent stem cell culture system. Stem cell research & therapy10(1), 1-16.

Line 449: The authors mention that there are only 32 studies describing reprogramming in farm animals. The citations provided are either review articles or articles which are relatively old (2019 and previous). In addition, one of the study cited [42] is a study on embryo derived pluripotent cells. Since 2019 several new studies describing livestock pluripotent cells have been reported. In the way this sentence is currently written, it is incorrect and does not reflect the current status of the field. Please add the current reports on livestock iPSCs or replace the older references with more updated ones. This is really important revision that needs to be made in this review article.

Line 383: Please include studies that have reported increased reprogramming efficiencies using species specific factors. eg: Nanog in bovine, SV40largeT in bovine, KDM4A in bovine, Lin28, SV40LT, hTERT in ovine, 

Sumer, H., Liu, J., Malaver-Ortega, L. F., Lim, M. L., Khodadadi, K., & Verma, P. J. (2011). NANOG is a key factor for induction of pluripotency in bovine adult fibroblasts. Journal of animal science89(9), 2708-2716.

Bao, et al, Reprogramming of ovine adult fibroblasts to pluripotency via drug-inducible expression of defined factors Cell Res. 2011 Apr; 21(4): 600–608.

Other minor revisions:

Line 191: Typo "eb" instead of "be"

Line 226, 355 and 373: Please keep style of  "in vitro" consistent

Line 445: Please rephrase the sentence. In the current state it is not easily understandable what the author is trying to convey.

Author Response

Reviewer 2 Questions and responses.

The review article describes the use of induced pluripotent reprogramming technology in conservation/preservation approaches and also the production of lab meat. The paper, in general, is well written with minor errors. However, there are some important corrections to be made to the review including the adding recent developments in the field before this is ready for publication.

Below are my comments:

  1. Line 345 ..."However, the re- 345 search on iPSC cultures of several domestic animals is still in its infancy". Please refrain from making such strong statements and please tone it down. It is true that domestic animal reprogramming advances have not reached the level to that of murine or primate, but in the recent years there has been several advances in domestic animal reprogramming technology. Please see references included below.

Su, Y., Wang, L., Fan, Z., Liu, Y., Zhu, J., Kaback, D., ... & Tang, Y. (2021). Establishment of Bovine-Induced Pluripotent Stem Cells. International journal of molecular sciences22(19), 10489.

Pillai, V. V., Koganti, P. P., Kei, T. G., Gurung, S., Butler, W. R., & Selvaraj, V. (2021). Efficient induction and sustenance of pluripotent stem cells from bovine somatic cells. Biology open10(10), bio058756.

Response: We thank reviewer for the suggestion, and we have amended in the manuscript accordingly. Please see line 407-408.

  1. Line 347:  Several papers published have indeed shown the potential of ipsc to differentiate into the 3 germ layers.  If germline contribution was what was intended, it is confusing to the reader as to what germ cell differentiation means. In short, if the author intended to convey was about contribution to germline, please rephrase the sentence.

Response: We thank the reviewer for the correction to be made in the manuscript as mentioned above. Therefore, we have amended accordingly on line 408-409.

  1. Line 348: The statement on use of small molecules on iPSC derivation in livestock has been examined and also been reported. Please correct the sentence and include the below references. 

Xu, J., Yu, L., Guo, J., Xiang, J., Zheng, Z., Gao, D., ... & Han, J. (2019). Generation of pig induced pluripotent stem cells using an extended pluripotent stem cell culture system. Stem cell research & therapy10(1), 1-16.

Response: We thank the reviewer for suggestion of reference to be added to the manuscript as marked. Therefore, we have included the reference and stated its content to the manuscript at line 411.

  1. Line 449: The authors mention that there are only 32 studies describing reprogramming in farm animals. The citations provided are either review articles or articles which are relatively old (2019 and previous). In addition, one of the study cited [42] is a study on embryo derived pluripotent cells. Since 2019 several new studies describing livestock pluripotent cells have been reported. In the way this sentence is currently written, it is incorrect and does not reflect the current status of the field. Please add the current reports on livestock iPSCs or replace the older references with more updated ones. This is really important revision that needs to be made in this review article.

Response: As suggested by the reviewer we have included new current references to the manuscript accordingly. Please see line 549-551

  1. Line 383: Please include studies that have reported increased reprogramming efficiencies using species specific factors. eg: Nanog in bovine, SV40largeT in bovine, KDM4A in bovine, Lin28, SV40LT, hTERT in ovine, 

Sumer, H., Liu, J., Malaver-Ortega, L. F., Lim, M. L., Khodadadi, K., & Verma, P. J. (2011). NANOG is a key factor for induction of pluripotency in bovine adult fibroblasts. Journal of animal science89(9), 2708-2716.

Bao, et al, Reprogramming of ovine adult fibroblasts to pluripotency via drug-inducible expression of defined factors Cell Res. 2011 Apr; 21(4): 600–608.

Response: As suggested by the reviewer regarding the references, we have included in our manuscript at line451-455

Other minor revisions:

  1. Line 191: Typo "eb" instead of "be"

Response: We have corrected as suggested at line247

  1. Line 226, 355 and 373: Please keep style of "in vitro" consistent

Response: We have made “in-vitro” as italics font throughout the manuscript as stated at Line:145,169,171,187,188,189,232,243,285,302,339,363,374,405,433,434,442,482,497,556

  1. Line 445: Please rephrase the sentence. In the current state it is not easily understandable what the author is trying to convey.

Response: We have rephrased the line in the manuscript as suggested by the reviewer. Please see correction at line 544-545.